# Maternal Pregnancy Hormone Concentrations in Countries with Very Low and High Breast Cancer Risk

**DOI:** 10.3390/ijerph17030823

**Published:** 2020-01-28

**Authors:** Davaasambuu Ganmaa, Davaasambuu Enkhmaa, Tsedmaa Baatar, Buyanjargal Uyanga, Garmaa Gantsetseg, Thomas T. Helde, Thomas F. McElrath, David E. Cantonwine, Gary Bradwin, Roni T. Falk, Robert N. Hoover, Rebecca Troisi

**Affiliations:** 1Channing Division Network of Medicine, Department of Medicine, Brigham and Women’s Hospital and Harvard Medical School and Department of Nutrition, Harvard T.H. Chan School of Public Health, Boston, MA 02115, USA; gdavaasa@hsph.harvard.edu; 2Mongolian Health Initiative, Ulaanbaatar 13312, Mongolia; uyanga.buyanjargal@gmail.com (B.U.); gantsetseg.garmaa@gmail.com (G.G.); 3Maternal and Child Health Research Center, Ulaanbaatar 16060, Mongolia; dv.enkhmaa@gmail.com (D.E.); tsedmaabaatar@gmail.com (T.B.); 4Information Management Services, Inc., Rockville, MD 20850, USA; th3j@nih.gov; 5Harvard Medical School, Department of Obstetrics and Gynecology, Brigham and Women’s Hospital, Boston, MA 02115, USA; tmcelrath@bwh.harvard.edu (T.F.M.); dcantonwine@bwh.harvard.edu (D.E.C.); 6Clinical and Epidemiologic Research Laboratory, Boston Children’s Hospital, Boston, MA 02115, USA; Gary.Bradwin@childrens.harvard.edu; 7Epidemiology and Biostatistics Program, Division of Epidemiology and Genetics, National Cancer Institute, Rockville, MD 20850, USA; falkr@mail.nih.gov (R.T.F.); hooverr@exchange.nih.gov (R.N.H.)

**Keywords:** pregnancy, hormones, early life, breast cancer, estrogen

## Abstract

Background: Breast cancer rates in Asia are much lower than in Europe and North America. Within Asia, rates are lower in Mongolia than in neighboring countries. Variation in pregnancy exposure to endogenous hormone concentrations may explain the differences, but data are lacking. Methods: We measured maternal serum progesterone, prolactin, estradiol and estrone concentrations in the second half of pregnancy in a cross-sectional study of urban (*n* = 143–194 depending on the analyte) and rural (*n* = 150–193) Mongolian women, and U.S. women from Boston (*n* = 66–204). Medical records provided information on maternal and perinatal factors. Geometric mean hormones were estimated from standard linear models with the log-hormone as the dependent variable and country as the independent variable adjusted for maternal and gestational age at blood draw. Results: Mean concentrations of prolactin (5722 vs. 4648 uIU/mL; *p* < 0.0001) and estradiol (17.7 vs. 13.6 ng/mL; *p* < 0.0001) were greater in Mongolian than U.S. women, while progesterone (147 vs. 201 ng/mL; *p* < 0.0001) was lower. Mean hormone concentrations were similar in rural and urban Mongolian women. Results were generally similar, with additional adjustment for gravidity, parity, height, body mass index at blood draw, education and alcohol use during pregnancy, and when stratified by offspring sex or parity. Conclusions: Mongolian women had greater concentrations of prolactin and estrogen and lower concentrations of progesterone than U.S. women, while hormone concentrations were similar in rural and urban Mongolian pregnancies. Impact: These data do not support the hypothesis that estrogen concentrations in pregnant women are lower in Mongolian compared with Caucasian women.

## 1. Introduction

One of the strongest risk factors for breast cancer remains the woman’s country of birth. Despite increases in breast cancer incidence in the developing world, rates in Northern and Western Europe and North America remain greater by approximately two to three times those of East and Southeast Asia [1]. Within Asia, rates in China and Japan are almost four to six times greater [1], respectively, than in neighboring Mongolia. These international differences appear largely due to environment, as incidence rates increase rapidly among Asian migrants to Western countries, reaching those of the host country by the second generation [2]. In Mongolia, the estimated breast cancer incidence rate, one of the lowest in the world (age-standardized rate in 2018: 11.3/100,000 vs. 84.9/100,000 in U.S. women) [1], is greater in women living in urban than rural areas [3]. Differences in established breast cancer risk factors, including endogenous hormone concentrations, have long been speculated to explain these epidemiological observations [4,5], but evidence is lacking. While there are some data indicating differences in steroid hormone concentrations between pre- and postmenopausal Asian and Caucasian populations [6,7], including from Mongolia [8], they are not adequate to explain the magnitude of the breast cancer incidence rate differences [9], perhaps indicating the importance of hormonal exposure during other critical periods of breast development, such as in pregnancy (for the mother) or prenatally (for the offspring). 

The reduction afforded by an early age at first birth and number of additional live births [10] indicates that exposures occurring in pregnancy play a critical role in the subsequent breast cancer risk of the mother. Several biological mechanisms have been offered to explain these observations [11] most prominently hormonally mediated molecular alterations in breast tissue that diminish susceptibility to carcinogenic transformation. The steroid hormones, estrogen and progesterone, and the pituitary peptide hormone, prolactin [12], are critical to the substantial epithelial cell proliferation in the breast occurring in pregnancy, making them key biomarker candidates. Several studies have now indicated associations between pregnancy hormones and breast cancer risk [13,14,15,16,17].

Prenatal exposure to hormones (in contrast to maternal pregnancy exposure) has also been speculated to affect breast cancer incidence based on increased risks associated with the steroid estrogen diethylstilbestrol [18], and 1,1,1-trichloro-2,2-bis(chlorophenyl)ethane (DDT) [19], a xenoestrogenic pesticide. Data on differences in pregnancy hormone profiles among international populations are extremely limited but provocative, showing *higher* estrogen and prolactin concentrations during pregnancy in Chinese women living in Beijing than in American women living in Boston [20], as well as elevations in several other pregnancy hormones [21]. We investigated variations in pregnancy hormones that are associated with subsequent breast cancer risk in rural and urban Mongolian women, Asian populations with even lower breast cancer incidence than the Chinese, and compared them with U.S. women who have a high incidence of breast cancer.

## 2. Materials and Methods 

### 2.1. Study Design

We conducted a cross-sectional study of maternal serum progesterone, prolactin, estradiol and estrone concentrations in the second half of pregnancy in urban and rural Mongolian women and in women living in Boston, MA, US.

### 2.2. Participant Identification and Recruitment

Eligible for study were pregnant women ≥18 years of age, who received prenatal care (not a referral) at the Maternity and Child Health Research Center Hospital or Bayangol Hospital in Ulaanbaatar (MCHRC), the Bulgan and Selenge general hospitals in Bulgan and Selenge provinces of rural Northern Mongolia, or Brigham and Women’s Hospital (BWH) in Boston, U.S., with singleton, naturally conceived (i.e., no use of artificial reproductive technologies), presumed to be viable pregnancies. 

In Mongolia, all eligible pregnant women were recruited during one of their second or third trimester routine visits (range 176–293 days; mean and median were 218 and 223 days for urban women, and 220 and 218 days for rural women). Samples were collected from September 2011 through June 2013. All women who were approached agreed to participate in the study: 197 women for the urban sample (MCHRC, Bayangol Hospital) and 196 women for the rural sample (Bulgan and Selenge hospitals). Three Mongolian women (1 in urban and 2 in rural Mongolia) were excluded from the analysis because of missing gestational age at blood draw, and another urban woman’s sample was excluded because it was collected too early in gestation. All 195 urban and 194 rural samples were submitted for the prolactin and progesterone assays. The estrogen assays, including 151 rural and 143 urban samples, were performed at a different time and excluded nulligravid women (i.e., women who had never been pregnant). 

A group from a Western country with high breast cancer incidence rates was chosen for comparison to the Mongolian group. Participants for the currently study were from LIFECODES, a pregnancy cohort followed by BWH [22]. Eligible for the pregnancy cohort were women who presented for routine prenatal care at ≤15 weeks’ gestation and planned to deliver at BWH. Samples were collected from November 2012 through December 2014. Of those eligible during the collection period, 46.5% agreed to participant in the birth cohort and had a maternal serum sample collected. Two hundred and seven blood samples from women meeting the eligibility criteria for the current study were collected sequentially at a routine third trimester visit (range 227–262 days; mean 246, median 245). All 207 were submitted for the prolactin and progesterone assays, while 66 nulligravid samples were submitted for the estrogen assays.

### 2.3. Biospecimen Collection, Processing and Storage

Two 10 mL red-top tubes of whole blood were drawn. At all sites, samples were immediately delivered to the hospital laboratory where they were left to clot at room temperature and then centrifuged. Serum samples collected in rural Mongolia were transported on ice to the laboratory in Ulaanbaatar and were stored together with the urban samples in freezers at the Health Sciences University of Mongolia until they were shipped to the U.S. National Cancer Institute (NCI) biorepository for long-term storage. The BWH samples were shipped to NCI’s biorepository at the end of the collection period. 

### 2.4. Laboratory Assays

Progesterone and prolactin assays were performed at the Clinical and Epidemiologic Research Laboratory at Boston Children’s Hospital (Boston, USA), and the estrogen assays were performed at the Frederick National Laboratory for Cancer Research, Frederick, Maryland. Equal proportions of samples from the three groups (rural Mongolia, urban Mongolia, U.S.) were randomly distributed in each assay batch. Blinded identical aliquots from a quality control pool of maternal sera were included with the study samples, constituting approximately 10% of each assay batch. The laboratory technicians also were blinded to country of sample origin. 

Progesterone was measured by a competitive electrochemiluminescence immunoassay on the Roche E Modular system (Roche Diagnostics, Indianapolis, IN). In short, a biotinylated progesterone antibody and a progesterone derivative labeled with ruthenium are mixed with the serum sample. The progesterone in the sample and the ruthenium-labeled progesterone compete for binding sites on the biotinylated antibody and form the respective immunocomplexes. Streptavidin-coated magnetic microparticles are then added to the reaction mixture to bind the biotinylated antibody. These immunocomplexes are magnetically entrapped on an electrode and the unbound reagents and sample are washed away. A chemiluminescent reaction is then electrically stimulated to generate light, the intensity being indirectly proportional to the amount of progesterone present in the sample. The lowest detection limit of this assay is 0.05 ng/mL and the day-to-day imprecision values at concentrations of 0.739, 8.96 and 54.6 ng/mL are 5.3, 2.9 and 2.4%, respectively. The overall (inter- and intra-batch) coefficient of variation (CV) for progesterone was 4.6%.

Prolactin was measured by a sandwich electrochemiluminescence immunoassay on the Roche E Modular system (Roche Diagnostics, Indianapolis, IN). In short, a biotinylated prolactin antibody and a prolactin antibody labelled with ruthenium are mixed with the serum sample. The prolactin in the sample is sandwiched between the ruthenium-labelled and biotinylated antibodies. Streptavidin-coated magnetic microparticles are then added to the reaction mixture to bind the biotinylated antibody. These immunocomplexes are magnetically entrapped on an electrode and the unbound reagents and sample are washed away. A chemiluminescent reaction is then electrically stimulated to generate light, the intensity being directly proportional to the amount of prolactin present in the sample. The lowest detection limit of this assay is 0.47 ng/mL and the day-to-day imprecision values at concentrations of 3.4, 30.9 and 109.6 ng/mL are 2.8, 2.5 and 3.4%, respectively. The overall CV for prolactin was 2.0%. 

Serum concentrations of estrone and estradiol were performed using stable isotope dilution liquid chromatography–tandem mass spectrometry (LC–MS/MS) at the Frederick National Laboratory for Cancer Research, Frederick, Maryland. Details of the method for measuring serum estrogens including sample preparation and assay conditions, have been published previously [23]. Mongolian and US-matched sets were assayed in the same batches, and blinded (to the laboratory) quality control samples based on a pool of extra serum from the study pregnancies constituted approximately 10% of each batch. No assays of estrogens resulted in non-detectable readings. Total within-batch CVs based on blinded replicates were 3.3% for estradiol and 3.1% for estrone. 

### 2.5. Other Information on the Woman and the Pregnancy 

Information on the index pregnancy (maternal age at blood draw, gestational week at blood draw, height, pre-pregnancy weight, weight at blood draw, offspring sex) was abstracted from the medical record. In Mongolia, participants were interviewed by trained study staff to ascertain information on marital status, education, parity, age at first full-term pregnancy, smoking and alcohol use during the index pregnancy, and history of pregnancy complications; this information was available from medical records for the Boston group. 

In Mongolia, weight and height at enrolment were measured by trained study personnel using standardized techniques, while in Boston, they were measured at the woman’s clinical visit. Body Mass Index (BMI) was formed by dividing weight by squared height. In Boston, gestational age was confirmed by first trimester ultrasound measurements. Preeclampsia in both groups was based on American College of Obstetrics and Gynecology (ACOG) and the International Society for the Study of Hypertension in Pregnancy (ISSHP) diagnostic criteria.

### 2.6. Statistical Analysis

The distribution of characteristics for the two populations were compared using t-tests and likelihood ratio chi-square statistics. Tertiles for height and weight were based on the distribution of both populations combined. Scatterplots of progesterone, prolactin and the estrogens by gestational age were examined. Analyte values were log-transformed to improve normality. Values were examined for outliers using residual values relative to the predicted value of the assay, based on a linear regression model with log-transformed hormone as the dependent variable and gestational age as the independent variable. Outliers were defined as 2 interquartile ranges above the 75% percentile or below the 25% percentile leading to 8 total exclusions for progesterone analyses (6 from the US, including 4 whose value exceeded the maximum detection level, and 2 from Mongolia), 10 for prolactin analyses (3 from the U.S. and 7 from Mongolia), 1 for estradiol (from Mongolia) and none for estrone. 

General linear models (GLM statement) using Statistical Analysis Systems (SAS) software version 9.4 (SAS Institute, Cary, North Carolina) with the log-transformed hormone as the dependent variable and country as the independent variable provided mean hormone differences by country. Models were adjusted for gestational day at blood draw (and a squared term in analyses of progesterone and prolactin) and maternal age (continuous variables). The coefficient, β, for country was exponentiated to estimate the geometric means for the groups (Mongolia and US, or rural and urban Mongolia). The percentage difference was defined as 100 × (exp(β)-1). In addition to gestational days at blood draw and maternal age, we further adjusted in individual models for parity (parous vs. nulliparous), height (highest tertile vs. lowest tertile; middle tertile vs. lowest tertile), BMI at blood draw (continuous variable), alcohol use during the index pregnancy (yes vs. no) and education (indicator variables for graduated high school/some college; graduated college vs. some high school). Additional models were restricted to samples collected between 32 and 37 weeks, were stratified by offspring sex and by parity, and excluded preeclamptics. 

### 2.7. Human Subjects

The study was approved by ethical review boards at Mongolia’s Ministry of Health, the National University of Mongolia, and in the United States (U.S.) at the National Cancer Institute (National Institutes of Health), and the Harvard School of Public Health and Brigham and Women’s Hospital in Boston. All participants provided written, informed consent.

## 3. Results

The three groups of pregnant women differed in most characteristics (Table 1). Mean (median) age of U.S. women was 34.1 (34.1) years, and 27.9 (27.0) and 28.7 (28.0), respectively, in urban and rural Mongolian women. U.S. women were less likely to be married than Mongolian women and had a higher level of formal education than rural Mongolian women. U.S. women were also taller and heavier, with a higher pre-pregnancy BMI and a greater increase in BMI during pregnancy. Urban Mongolian women were more educated than rural women. Urban women were also taller than rural women and had a lower pre-pregnancy BMI but a greater increase in BMI during pregnancy. The number of prior pregnancies (gravidity) was similar (for progesterone and prolactin), but parity was greater in the rural Mongolian women than in the other two groups of women. The proportion of women who reported smoking during pregnancy was low in all three groups. Alcohol use was also low overall in pregnancy but higher in U.S. women and rural Mongolian women compared with urban Mongolian women. There were no differences among groups in offspring sex. The differences in characteristics between the Mongolian and U.S. women in the estrogen sample were similar although of smaller magnitude, and all women in both groups were gravid with no difference in number of prior pregnancies (data not shown). The racial/ethnic breakdown of the U.S. women was as follows: 64.7% white, 11.0% African-American, 6.8% Asian, 15.5% Hispanic and 2.4% other.

Mean progesterone, prolactin and estrogen concentrations adjusted for maternal and gestational age differed between Mongolian and U.S. women (Table 2) but were similar between rural and urban Mongolian women. Mean prolactin was 21% greater (*p* < 0.0001) and mean estradiol was 27% greater (*p* < 0.0001) in Mongolian women than U.S. women, while mean progesterone was 19% lower (*p* < 0.0001). Estrone concentrations did not differ between Mongolian and U.S. women, and differences in all hormones between pregnant rural and urban Mongolian women were < 10%.

With adjustment for parity, the lower progesterone concentration in Mongolian women (−19%) compared with U.S. women was attenuated slightly (−14%; *p* = 0.0008). None of the other hormone differences between Mongolian and U.S. women, or rural and urban Mongolian women were affected by adjustment for gravidity (progesterone and prolactin only as all women in the estrogen sample were gravid), height, BMI at blood draw, alcohol use during the index pregnancy and education (data not shown). The differences between Mongolian and U.S. pregnancies were generally similar when analyses were stratified by offspring sex and parity (data not shown).

In sensitivity analyses, differences between the Mongolian and U.S. women were similar to the overall results with gestational age at blood draw restricted to 32–37 weeks (progesterone −22%, prolactin 26%, estradiol 30%, estrone 9%, respectively), and when preeclampsia cases (*n* = 6 in Mongolia and *n* = 9 in Boston) were excluded (progesterone −19%, prolactin 21%, estradiol 28%, estrone −3%, respectively). 

## 4. Discussion

Direct measurements of the prenatal environment in markedly different breast cancer risk populations are rare. Our overall results for several hormones with established roles in breast carcinogenesis show differences between the maternal circulation in pregnancies occurring in Mongolia, with one of the lowest breast cancer incidence rates in the world, and in the US, with one of the highest incidence rates. Data from another Asian population are consistent with our results. A previous study of pregnant women living in Shanghai, China and Boston, U.S. found that circulating pregnancy concentrations of estradiol (23% compared with 27% in our study) and prolactin (27% vs. 21% in our study) were higher in the Asian women at gestational week 27, while progesterone concentrations (6.3% vs. 19% in our study) were lower [20]. In contrast, we found mostly no difference in estrone concentrations in Mongolian compared with U.S. women, and our results were generally similar with further adjustment for gravidity, height, BMI at blood draw, education and alcohol use, while adjustment for parity attenuated differences in progesterone. Findings from our prior study of premenopausal women are consistent with higher estrogen concentrations in Mongolian women than women living in the west [8]. Mean estradiol concentrations were almost 20% greater in Mongolian than British women, adjusted for age, parity, BMI, height, and smoking status. In contrast to lower levels observed in Mongolian than U.S. pregnancies, progesterone was approximately 50% higher in *premenopausal* Mongolian than British women, particularly during the follicular phase and early luteal surge, and there was a decreasing progesterone trend by degree of westernization (rural Mongolia; urban Mongolia; U.K.). Differences in environmental factors, in particular, diet, may be responsible for the higher estrogen and prolactin and lower progesterone concentrations in Mongolian mothers. 

The differences in pregnancy hormones between a high- and low-incidence population in our study are inconsistent with lower estrogen exposure in pregnancy providing a breast cancer risk benefit, instead suggesting that *higher* levels are protective, at least during pregnancy. Indeed, studies in animal models show reductions in breast tumor incidence with administration of estrogen and progesterone to mimic pregnancy [11]. Also, epidemiologic data show inverse associations for higher circulating levels of estradiol and estrone in the mother during pregnancy and her subsequent breast cancer risk when diagnosed at 40 years of age or older, but positive associations when diagnosed younger and closer to the pregnancy [13,14,15,16,17]. Since the international difference in breast cancer rates is most pronounced in older women [24], the higher pregnancy estrogen concentrations in Mongolian than U.S. women could be important. Lipworth et al. [20] posited several mechanisms for a pregnancy estrogen effect on breast cancer risk including down-regulation of estrogen receptor expression in breast tissue throughout life, possibly epigenetically mediated, increased mammary cell differentiation (lessening susceptibility to carcinogenesis) from higher levels of pregnancy estrogens interacting with other mammotrophic hormones, and differences in estrogen bioavailability.

The lower progesterone concentrations that we observed in Mongolian women are not consistent with studies showing an inverse association of maternal progesterone with breast cancer risk in younger women and no association with risk in older women at diagnosis [15,16,17]. Epidemiologic studies are lacking on pregnancy prolactin concentrations and breast cancer risk [25], but the higher levels in Mongolian women could indicate a possible protective effect for breast cancer.

Over the last decade and a half, Mongolia has experienced profound economic changes, resulting in mass migration from a nomadic or semi-nomadic existence to a more Western lifestyle in the capital city of Ulaanbaatar. Together with the contrast in exposures between traditional and urban settings, migration presents the opportunity to study women as they acculturate to a more Western lifestyle. With migration, breast cancer incidence may have increased in women now living in urban Mongolia [3]. We found, however, that maternal pregnancy hormone concentrations were similar in the rural and urban Mongolian women indicating that environmental factors that influence the prenatal hormonal milieu have not changed with migration from rural to urban areas or that the hormones measured do not explain the difference in breast cancer rates. 

Whether generalizing our results to Asian and Western populations is valid depends on the representativeness of the samples of women we recruited. The U.S. is multiethnic and genetically more diverse than Mongolia. The racial/ethnic breakdown of the women recruited in Boston was like that for the general U.S. population according to the 2010 census [26]. By not limiting participants to Caucasians, we measured hormone values in a sample of women from which the U.S. breast cancer rates arose. Health care, including pre-natal and delivery, are free in Mongolia because of their national healthcare system, so the women who were recruited were unlikely to differ by socioeconomic status. At all sites, women were required to have had prenatal care at the hospital where they were recruited, and we excluded women who were referred for high-risk pregnancies. Women may have chosen to attend these hospitals for prenatal care because of a history of difficult pregnancies, but this would be true for hospitals in Boston and Ulaanbaatar. Results were similar when the small number of preeclampsia cases were removed from analysis. Less than half of eligible women in the Boston cohort agreed to provide a blood sample. If the reason for not participating were associated with hormone concentrations, then this would affect the generalization of our findings for these women.

We attempted to standardize blood collection, processing and storage protocols for all three sites but the possibility of differences in handling could explain some of the differences we observed. It is reassuring that differences in the hormones varied in direction (i.e., higher in Mongolian women for estradiol, estrone and prolactin but lower in progesterone) and were consistent with a previous study of Chinese and Boston women [20]. Another possibility, that reduced plasma volume expansion could be responsible for higher concentrations in the Mongolian women who were of smaller body size, is contradicted by their lower progesterone values compared with U.S. women. We also tried to minimize this potential bias by adjusting the hormones for BMI. The lack of ultrasound in the Mongolian pregnancies could have resulted in greater misclassification of gestational age than in the U.S. pregnancies. A large study of gestational age based on last menstrual period (LMP) found little difference in estimates of gestational age (SD 0.8 days; *median* = 0), on average than with ultrasound [27], and any error should be random rather than systematically biased toward greater gestational age. Because our sample for the estrogen analyses included excluded nulligravid women, our results for estrone and estradiol can only be generalized to gravid women. Performing data collection in this setting was difficult for a variety of logistical reasons making serial measurements not possible. Our findings may thus only pertain to hormone concentrations in later pregnancy, but existing data suggest that hormones measured in the different trimesters are highly correlated and track over the pregnancy [28].

In summary, estrogen’s influence on breast carcinogenesis has been known for several decades. International differences in breast cancer incidence rates have been speculated to be explained by less estrogen exposure in lower risk populations. Our data and those of others [20] suggest, however, that greater estrogen levels in pregnancy may be beneficial for the long-term risk of breast cancer. While we have ample reason to believe that exposures that occur early in life, including those in pregnancy, play a role in breast cancer development, a comprehensive etiologic understanding is still lacking. The descriptive data collected in the current study provide a way to test our current hypotheses. However, the biology of pregnancy associated changes in the breast and how they affect cancer risk is undoubtedly complicated and requires further study.

## Figures and Tables

**Table 1 ijerph-17-00823-t001:** Number (%) of Rural and Urban Mongolian and U.S. Women by Demographic, Anthropometric and Pregnancy Characteristics.

	Mongolia			U.S.	
	Rural	Urban	p ^a^		p ^b^
	*n* = 194	*n* = 195		*n* = 207	
Age (years)			0.27		<0.0001
<20	4 (2.1)	3 (1.5)		2 (0.97)	
20–24	52 (26.8)	56 (28.7)		9 (4.4)	
25–29	57 (29.4)	75 (38.5)		32 (15.5)	
30–34	44 (22.7)	34 (17.4)		72 (34.8)	
35–39	30 (15.5)	24 (12.3)		62 (30.0)	
≥40	7 (3.6)	3 (1.5)		30 (14.5)	
Gestational weeks at blood draw			0.01		<0.0001
25.1 < 31.5	108 (55.7)	90 (46.2)		0 (0.0)	
31.5–34.3	65 (33.5)	93 (47.7)		39 (18.8)	
34.4–41.9	21 (10.8)	12 (6.2)		168 (81.2)	
Marital status			0.72 ^c^		<0.0001
Single	4 (2.0)	2 (1.0)		30 (14.5)	
Married	189 (97.4)	192 (98.5)		170 (82.1)	
Divorced/separated/widowed	1 (0.5)	1 (0.5)		7 (3.4)	
Education			<0.0001		<0.0001
<High School (HS)	90 (46.4)	0 (0.0)		2 (1.0)	
HS or some college	90 (46.4)	73 (37.9)		53 (25.6)	
≥College	14 (7.2)	122 (62.1)		152 (73.4)	
Gravidity (before current pregnancy)			0.21		0.44
0	41 (21.1)	51 (26.2)		39 (18.8)	
1	52 (26.8)	48 (24.6)		58 (28.0)	
2	36 (18.6)	46 (23.6)		53 (25.6)	
3	29 (15.0)	28 (14.4)		32 (15.5)	
≥4	36 (18.6)	22 (11.3)		25 (12.1)	
Parity (before current pregnancy)			<0.0001		0.004
0	51 (26.3)	85 (43.6)		90 (43.7)	
1	63 (32.5)	73 (37.4)		79 (38.4)	
2	80 (41.2)	37 (19.0)		37 (18.0)	
Missing	0 (0.0)	0 (0.0)		1 (0.5)	
Height (tertiles; cm)			0.002		<0.0001
<157.5	90 (46.6)	56 (28.7)		52 (25.5)	
157.5–164.0	65 (33.7)	87 (44.6)		44 (21.6)	
>164.0	38 (19.7)	52 (26.7)		108 (52.9)	
	0 (0.0)	0 (0.0)		3 (1.5)	
Prepregnancy weight (tertiles; kg)			0.04		<0.0001
<55.33	73 (37.6)	83 (42.6)		38 (19.1)	
55.34–64.9	63 (32.5)	67 (34.4)		60 (30.2)	
≥65.0	53 (27.3)	44 (22.6)		101 (50.8)	
missing	5 (2.6)	1 (0.5)		8 (3.9)	
Prepregnancy BMI (wt/ht^2^)			0.01		<0.0001
<20	25 (12.9)	48 (24.6)		24 (11.9)	
20–25	112 (57.7)	103 (52.8)		86 (42.6)	
25–29	41 (21.1)	34 (17.4)		51 (25.3)	
>30	10 (5.2)	10 (4.6)		40 (20.3)	
Missing	6 (3.1)	0 (0.0)		5 (2.4)	
BMI at blood draw (wt/ht^2^)			0.36		<0.0001
<25	47 (24.2)	55 (28.2)		32 (15.7)	
25–39	96 (49.5)	100 (51.3)		71 (34.8)	
≥30	50 (25.8)	40 (20.5)		101 (49.5)	
Missing	1 (0.5)	0 (0.0)		3 (1.4)	
Δ BMI ^d^ (wt/ht^2^)			0.006		0.03
<3.71	81 (41.8)	56 (28.7)		57 (28.6)	
3.71–5.31	61 (31.4)	73 (37.4)		61 (30.7)	
>5.31	46 (23.7)	65 (33.3)		81 (40.7)	
Missing	6 (3.1)	1 (0.5)		8 (3.9)	
Smoking during pregnancy			1.0 ^c^		<0.0001
No	194 (100.0)	193 (99.0)		194 (94.2)	
Yes	0 (0.0)	1 (0.5)		12 (5.8)	
Missing	0 (0.0)	1 (0.5)		0 (0.0)	
Alcohol use during pregnancy			0.001 ^c^		0.04
No	177 (91.2)	192 (98.5)		187 (90.3)	
Yes	17 (8.8)	3 (1.5)		20 (9.7)	
Sex of offspring			0.21		0.38
Female	84 (43.3)	98 (50.3)		104 (50.7)	
Male	109 (56.2)	97 (49.7)		101 (49.3)	
Missing	1 (0.5)	0 (0.0)		2 (1.0)	

Percentages do not always sum to 100 because of rounding. ^a^ p-value for rural vs. urban. ^b^ p-value for U.S. vs. Mongolia (combined urban and rural). ^c^ increase in BMI from pre-pregnancy to blood draw adjusted for gestational week at blood draw; women who lost weight are in the lowest tertile category. ^d^ p-value is based on Fisher’s exact test.

**Table 2 ijerph-17-00823-t002:** Hormone Concentrations ^a^ in Women Living in Rural and Urban Mongolia and Boston, U.S.

Hormone	Rural Mongolia			Urban Mongolia			Δ ^b^	Total Mongolia			Boston, U.S.			Δ ^b^
	n	Mean	95% CI	n	Mean	95% CI		n	Mean	95% CI	n	Mean	95% CI	
Progesterone ng/mL	193	134	128–140	194	130	125–136	3%	387	147	141–152	201	177	166–188	−19%
Prolactin uIU/mL	188	5509	5198–5838	194	5763	5442–6102	−5%	382	5722	5448–6010	204	4648	4313–5009	21%
Estradiol ng/mL	150	17.4	16.4–18.4	143	17.2	16.2–18.3	1%	293	17.7	17.0–18.5	66	13.6	12.2–15.1	27%
Estrone ng/mL	151	50.1	45.1–55.8	143	45.9	41.1–51.2	9%	294	49.8	46.0–54.0	66	50.9	41.8–62.0	−2%

^a^ Geometric means adjusted for maternal age and gestational age at sampling (models for progesterone and prolactin also include gestational age*gestational age). ^b^ The p-values were < 0.0001 for all hormones comparing Mongolia to U.S. except for estrone (*p* = 0.85). The p-values for the rural vs. urban Mongolia comparison were 0.35 for progesterone, 0.28 for prolactin, 0.80 for estradiol, and 0.25 for estrone. When %Δ is positive, values are higher in Mongolia than U.S. (and in rural than urban).

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
