# Peer review of "Maternal Pregnancy Hormone Concentrations in Countries with Very Low and High Breast Cancer Risk"

_ijerph, 2020, doi:10.3390/ijerph17030823_

Round 1

Reviewer 1 Report

The authors should be commended for the field work that was conducted. The goal of this study was to compare differences in maternal pregnancy estrogen concentrations in two groups: Mongolia and Boston. Prior literature suggests that estrogen levels during pregnancy may contribute to breast cancer risk later in life. This was a well-conducted cross-sectional study, but given the design, the Discussion should include more cautious interpretation. Other comments are listed below.

Abstract needs to state study design and sample sizes. The abstract’s conclusion needs to be re-stated because breast cancer risk was not specifically assessed in this cross-sectional study. Further, the study only focused on Mongolian women, so the sentence may not be generalizable to all Asians.

Introduction: the role of diet is not mentioned in the incidence of breast cancer, and these vary by country. In Mongolia, no information is presented regarding a national tumor registry, so it is unclear if the rates are low due to lack of mechanism to report cancer cases, and if the denominator is accurate. The actual breast cancer incidence of the two countries are not stated.

Methods: Whether all women in the Mongolian provinces delivered at the study hospitals; there may be selection bias with more women with means being able to deliver at hospitals. For the Boston cohort, there was no mention of race/ethnicity, which is an important risk factor. What were the median storage times for the samples from Mongolia and Boston, and could this have affected the lab assay results?

Discussion: the inference regarding the possible biologic mechanism how high estrogen levels during pregnancy may affect (lower) later breast cancer risk are interesting. However, women with breast cancer are recommended to take drugs that lower estrogen to reduce their risk of developing recurrence. Thus, the two mechanisms appear to conflict.

Limitations of the current study need to be stated such as lack of diet information, and other potential confounders. The discussion (last paragraph) should mention how many prior studies found greater levels of estrogen were protective, and the level of rigor of these prior studies in terms of weaknesses – were there any longitudinal retrospective studies published that support the study’s inference?

It was not clear why serial measurements (longitudinal over a time window of the pregnancy) could not have been done. The longitudinal data would have bolstered the inferences. The main important  limitation is the lack of repeated measurements over time.

Author Response

Dear Reviewer,

Thank you for your careful review and helpful comments. Please see below for our response

Reviewer #1

The authors should be commended for the field work that was conducted. The goal of this study was to compare differences in maternal pregnancy estrogen concentrations in two groups: Mongolia and Boston. Prior literature suggests that estrogen levels during pregnancy may contribute to breast cancer risk later in life. This was a well-conducted cross-sectional study, but given the design, the Discussion should include more cautious interpretation. Other comments are listed below.

We thank the reviewer for comments about the study. With regard to the Discussion, we believe  we were reasonable in our interpretation of the data and put it in context with the results of other studies like ours and other types of studies. In addition, we discussed several potential limitations. Reflecting this, in the concluding paragraph we wrote: “While we have ample reason to believe that exposures that occur early in life, including those in pregnancy, play a role in breast cancer development, a comprehensive etiologic understanding is still lacking. The descriptive data collected in the current study provides a way to test our current hypotheses, however, the biology of pregnancy associated changes in the breast and how they affect cancer risk is undoubtedly complicated and requires further study.”

Abstract needs to state study design and sample sizes. The abstract’s conclusion needs to be re-stated because breast cancer risk was not specifically assessed in this cross-sectional study. Further, the study only focused on Mongolian women, so the sentence may not be generalizable to all Asians.

We have stated that the study is cross-sectional to the Abstract and added the sample sizes. We have revised the last sentence of the Abstract to: “Impact: These data do not support the hypothesis that estrogen concentrations in pregnant are lower in Mongolians compared with Caucasian women.”

Introduction: the role of diet is not mentioned in the incidence of breast cancer, and these vary by country. In Mongolia, no information is presented regarding a national tumor registry, so it is unclear if the rates are low due to lack of mechanism to report cancer cases, and if the denominator is accurate. The actual breast cancer incidence of the two countries are not stated.

In our analysis we were specifically interested in hormone concentrations. Dietary intake as well as other potential risk factors do vary by country and any differences that we found could be explained by different diets in the populations. We have added this to the end of the first paragraph in the Discussion. “Differences in environmental factors, in particular, diet, may be responsible for the higher estrogen and prolactin and lower progesterone concentrations in Mongolian mothers.”

Regarding breast cancer incidence rates in Mongolia, we cited our paper (Troisi R, Altantsetseg D, Davaasambuu G, Rich-Edwards J, Davaalkham D, Tretli S, Hoover RN, Frazier AL. Breast cancer incidence in Mongolia. Cancer Causes Control 2012;23:1047-53.) which describes the tumor registry and its potential limitations. We have now added incidence rates to the first paragraph of the Introduction.

Methods: Whether all women in the Mongolian provinces delivered at the study hospitals; there may be selection bias with more women with means being able to deliver at hospitals. For the Boston cohort, there was no mention of race/ethnicity, which is an important risk factor. What were the median storage times for the samples from Mongolia and Boston, and could this have affected the lab assay results?

Health care, including pre-natal and delivery, are free in Mongolia because of their national healthcare system. We have added this to the 5th paragraph of the Discussion: “Health care, including pre-natal and delivery, are free in Mongolia because of their national healthcare system, so the women who were recruited were unlikely to differ by socioeconomic status.”

The racial/ethnic breakdown of the Boston cohort was stated at the end of the first paragraph of the results: “The racial/ethnic breakdown of the US women was as follows: 64.7% white, 11.0% African-American, 6.8% Asian, 15.5% Hispanic and 2.4% other.” Also, it was referred to in the Discussion: “The US is multiethnic and genetically more diverse than Mongolia. The racial/ethnic breakdown of the women recruited in Boston was like that for the general US population according to the 2010 census.26

We made every effort to standardize the blood processing at the two sites, but it is possible that median storage time differed slightly. We stated this in the discussion of limitations in the 6th paragraph of the Discussion. “We attempted to standardize blood collection, processing and storage protocols for all three sites but the possibility of differences in handling could explain some of the differences we observed. It is reassuring that differences in the hormones varied in direction (i.e. higher in Mongolians for estradiol, estrone and prolactin but lower in progesterone) and were consistent with a previous study of Chinese and Boston women.20

Discussion: the inference regarding the possible biologic mechanism how high estrogen levels during pregnancy may affect (lower) later breast cancer risk are interesting. However, women with breast cancer are recommended to take drugs that lower estrogen to reduce their risk of developing recurrence. Thus, the two mechanisms appear to conflict.

We attempted to clarify that we are referring to estrogen concentrations in pregnancy as is Lipworth’s biological hypotheses that we reference. “Lipworth et al.20 posited several mechanisms for a pregnancy estrogen effect on breast cancer risk including down-regulation of estrogen receptor expression in breast tissue throughout life, possibly epigenetically mediated, increased mammary cell differentiation (lessening susceptibility to carcinogenesis) from higher levels of pregnancy estrogens interacting with other mammotrophic hormones, and differences in estrogen bioavailability.”

Limitations of the current study need to be stated such as lack of diet information, and other potential confounders. The discussion (last paragraph) should mention how many prior studies found greater levels of estrogen were protective, and the level of rigor of these prior studies in terms of weaknesses – were there any longitudinal retrospective studies published that support the study’s inference?

As mentioned in the response above, we were interested in studying whether data were consistent with the hypothesis that hormone concentrations may be involved in differences in breast cancer rates. If dietary intake affects hormone concentrations, then it would not be appropriate to adjust for them because they are earlier in the biological pathway. We have added this to the end of the first paragraph in the Discussion. “Differences in environmental factors, in particular, diet, may be responsible for the higher estrogen and prolactin and lower progesterone concentrations in Mongolian mothers.”

It was not clear why serial measurements (longitudinal over a time window of the pregnancy) could not have been done. The longitudinal data would have bolstered the inferences. The main important  limitation is the lack of repeated measurements over time.

Performing data collection in this setting was difficult for a variety of logistical reasons making serial measurements not possible. The reviewer makes an important point about the timing of the blood collection as our findings may only pertain to hormone concentrations in later pregnancy. The existing data suggest, however, that hormones measured in the different trimesters are highly correlated and track over the pregnancy (Schock HZeleniuch-Jacquotte ALundin EGrankvist KLakso HÅIdahl ALehtinen MSurcel HMFortner RT. Hormone concentrations throughout uncomplicated pregnancies: a longitudinal study. BMC Pregnancy Childbirth 2016;16:146.). We have added this to the discussion of limitations in the Discussion.

Reviewer 2 Report

This manuscript compared the hormone levels in pregnancies between Mongolians with low breast cancer incidence and U.S. women with a high incidence of breast cancer and tried to explain the variations in risk of breast cancer. It would be interesting for readers. However, before publication, there were a few issues to be addressed.

Only 46.5% of eligible subjects agreed to give serum samples, then how was the representativeness? Table 2 showed that the total means of progesterone and estradiol of women were higher than both individual means form rural Mongolia and urban Mongolia. Is that reasonable? More discussion may be needed, particularly for the hormone levels differences between pre/post-menopausal women in high and low risk areas. The forms of the Tables were not academically correct.

Author Response

Dear Reviewer,

Thank you for your careful review and helpful comments. Please see below for our response:

This manuscript compared the hormone levels in pregnancies between Mongolians with low breast cancer incidence and U.S. women with a high incidence of breast cancer and tried to explain the variations in risk of breast cancer. It would be interesting for readers. However, before publication, there were a few issues to be addressed.

Only 46.5% of eligible subjects agreed to give serum samples, then how was the representativeness? Table 2 showed that the total means of progesterone and estradiol of women were higher than both individual means form rural Mongolia and urban Mongolia. Is that reasonable? More discussion may be needed, particularly for the hormone levels differences between pre/post-menopausal women in high and low risk areas. The forms of the Tables were not academically correct. 

We have added this to the 5th paragraph of the Discussion: “Less than half of eligible women in the Boston cohort agreed to provide a blood sample. If the reason for not participating were associated with hormone concentrations, then this would affect the generalization of our findings for these women.”

The discrepancy in means that the reviewer noticed can happen when geometric means are output from separate models. In this case, we performed one model comparing all Mongolians to US women and in a second model we compared rural and urban Mongolians only. The unadjusted geometric means for all Mongolians are as expected, a weighted average of the rural and urban women:

Progesterone: Total 132.0, Urban 127.8, Rural 136.9

Estradiol: Total 17.3, Urban 17.2, Rural 17.5

We are not sure how the Tables were not formatted correctly but are happy to revise them if the Editor can clarify what is required.